# TOWARDS BRINGING ADVANCED RESTORATION NETWORKS INTO SELF-SUPERVISED IMAGE DENOISING

## ABSTRACT

Self-supervised image denoising (SSID) has witnessed significant progress in recent years. Therein, most methods focus on exploring blind-spot techniques while only employing a simple network architecture (*e.g.*, plain CNN or U-Net) as a denoising backbone. However, with the ongoing advancements in image restoration networks, these architectures have become somewhat outdated. In this work, we aim to migrate the advanced restoration network designs (*e.g.*, SwinIR, Restormer, NAFNet, and HAT) into SSID methods. We begin by conducting an analysis of the fundamental concepts in existing typical blind-spot networks (BSN). Subsequently, we introduce a series of approaches to adapt restoration networks into various blind-spot ones. In particular, we suggest effective adjustment for window attention to mimic the convolution layers in BSN. And we discourage the adoption of channel attention in multi-level architectures, as it can potentially lead to the leakage of blind-spot information, consequently impeding performance. Experiments on both synthetic and real-world RGB noisy images demonstrate our methods substantially enhance SSID performance. Furthermore, we hope this study could enable SIDD methods to keep pace with the progress in restoration networks, and serve as benchmarks for future works. The code and pre-trained models will be publicly available.

## 1 INTRODUCTION

Image denoising is a fundamental low-level vision task that aims to recover clean images from noisy observations. With the development of neural networks, learning-based methods (Mao et al., 2016; Zhang et al., 2017a; Tai et al., 2017; Zhang et al., 2018a) have shown significant improvement against traditional patch-based ones (Buades et al., 2005; Dabov et al., 2007; Gu et al., 2014). Most of the early works synthesize noisy images with additive white Gaussian noise (AWGN) for training models. Since the noise distribution gap, these models exhibit degraded performance in real-world scenarios. One feasible solution is to capture noisy-clean image pairs (Plotz & Roth, 2017; Abdelhamed et al., 2018), and take them to train denoising models (Guo et al., 2019; Anwar & Barnes, 2019; Kim et al., 2020). However, the data collection process requires a rigorously controlled environment and much human labor.

Recently, self-supervised image denoising (SSID) methods have been introduced to circumvent the requirement of the paired dataset. Noise2Void (Krull et al., 2019) proposes a blind-spot technique that splits the noisy images into input and target parts with a masking strategy, thus being able to train networks with noisy images only. Instead of using the mask input strategy, some works (Cha & Moon, 2019; Laine et al., 2019; Wu et al., 2020; Byun et al., 2021) design dedicated modules (*i.e.*, blind-spot networks, BSN) that can exclude the corresponding input pixel from the receptive field of each output location. Moreover, probabilistic inference (Laine et al., 2019) and regularization loss functions (Huang et al., 2021; Wang et al., 2022a) are proposed to address the information loss problem at blind-spot, and they mitigate the performance gap between SSID and the supervised methods on synthetic noise. However, BSN is generally developed under the assumption of pixel-wise independent noise. In order to remove spatially correlated noise in real-world scenarios, it is suggested to break the noise correlation with pixel-shuffle downsampling (PD) (Wu et al., 2020; Zhou et al., 2020). And asymmetric PD factors during training and inference have shown better trade-off between noise removal and detail preserving (Lee et al., 2022; Wang et al., 2023).

Despite the rapid development, the most common neural architectures used in SSID methods are plain convolutional neural networks (CNN) or multi-level U-Net (Ronneberger et al., 2015), which falls largely behind the network design in image restoration. For example, window attention (Liu et al., 2021) exhibits strong local fitting capability and is successfully employed in some image restoration works (Liang et al., 2021; Wang et al., 2022b). Restormer (Zamir et al., 2022) performs self-attention along channel dimension. More recently, HAT (Chen et al., 2023) suggests a hybrid architecture that combines both channel attention and window attention. NAFNet (Chen et al., 2022) proposes a simple yet effective network design without nonlinear activations. In comparison, only a few attempts (Wang et al., 2023; Papkov & Chizhov, 2023) have been made to apply advanced network architectures to SSID, while the performance improvement is still unsatisfactory.

In this paper, we aim to migrate the advanced neural architecture design in image restoration into SSID methods. And we start by revisiting the building concepts of some typical blind spot networks. As shown in Fig. 1, they can be divided into two categories according to the construction manner. Firstly, FC-AIDE (Cha & Moon, 2019) and Laine19 (Laine et al., 2019) are conducted with multiple network branches, each of which has its receptive field restricted to a specific direction. At the end of the branches, the features are connected to form a full receptive field except for the blind-spot. Secondly, DBSN (Wu et al., 2020) and FBINet (Byun et al., 2021) are conducted with masked and dilated convolutions. The receptive field of each layer is particularly designed, thus maintaining the blind-spot mechanism throughout the whole network. Based on the above concepts, more advanced neural architectures have the potential to be adapted into blind-spot ones to improve SSID.

Specifically, we introduce a series of approaches for integrating various blind-spot manners and cutting-edge architecture designs effectively. In particular, we focus on the adjustments of the attention mechanism that is widely used in advanced restoration networks. On the one hand, for window attention, we suggest a specific mask adding to the attention weights. The mask discards weights at certain positions according to the relative spatial locations, thus creating a partial or sparse receptive field for building the blind-spot. On the other hand, for channel attention, we discourage its utilization in multi-level architectures in SSID. We find that, in deep layers of this architecture, as the channel number increases and spatial size decreases, spatial information tends to be shuffled into the channel dimension. The interaction between channels may leak spatial information at the blind-spot, leading to overfitting to noise.

Experiments are conducted on both synthetic and real-world RGB noisy images. We incorporate four representative and well-known image restoration networks (*i.e.*, SwinIR, Restormer, NAFNet, and HAT) into various blind-spot ones. All the advanced architectures consistently exhibit improvements while maintaining similar computational costs compared to the baselines, which demonstrates their effectiveness for SSID. Furthermore, some of them show favorable performance against state-of-the-art methods. Besides, we hope our work could bring attention to the development of more advanced BSN architectures, and serve as a foundational reference for future works.

The contributions of this study can be summarised as follows:

- We notice the lack of research on backbone architectures in self-supervised image denoising (SSID), and suggest adapting the advanced designs in restoration networks into SSID.
- We propose a series of adaptation approaches. Especially, we suggest an effective adjustment for window attention with attention masks, and discourage the adoption of channel attention in multi-level architectures.
- Experiments on multiple restoration networks migrating to various blind-spot networks show that, the proposed method achieves better performance than the state-of-the-art ones.

## 2 RELATED WORK

### 2.1 DEEP IMAGE DENOISING

The development of learning based methods (Zhang et al., 2017a; 2018a) has shown superior performance against traditional patch-based ones (Buades et al., 2005; Dabov et al., 2007; Gu et al., 2014) on Gaussian denoising. More advanced deep neural architectures (Mao et al., 2016; Tai et al., 2017; Liu et al., 2018) are further proposed to improve the learning ability. However, due to the domain gap between synthetic and real noise, models trained on Gaussian noise exhibit little denoising effect

on real-world noisy images (Plotz & Roth, 2017). In order to mitigate the noise discrepancy, (Guo et al., 2019; Zamir et al., 2020) suggest simulating more realistic noise, while (Abdelhamed et al., 2018) collects real-world pairs to train networks. With the help of these training data, denoising methods (Anwar & Barnes, 2019; Yue et al., 2019; Kim et al., 2020; Yue et al., 2020; Cheng et al., 2021; Ren et al., 2021) for real-world noisy images are rapidly proposed. Nonetheless, the noise statistics vary in different camera sensors and illuminating conditions (Wei et al., 2020; Zhang et al., 2021a), it is less practical to collect paired datasets for every device and scenario.

## 2.2 Self-Supervised Image Denoising

To circumvent collection of the paired data, self-supervised image denoising (SSID) approaches seek to utilize information from the noisy images themselves as supervision (Lehtinen et al., 2018; Krull et al., 2019; Batson & Royer, 2019). In order to prevent trivial solutions such as over-fitting to the identity mapping, blind-spot networks (BSN) (Cha & Moon, 2019; Laine et al., 2019; Wu et al., 2020; Byun et al., 2021) exclude the corresponding noisy pixel from the receptive field at every spatial location. Probabilistic inference (Laine et al., 2019; Krull et al., 2020) and regular loss functions (Huang et al., 2021; Wang et al., 2022a) are also proposed to leverage the information in the blind-spot. Recently, SwinIA (Papkov & Chizhov, 2023) explores transformer-based architectures (Liu et al., 2021; Liang et al., 2021) in SSID, but shows inferior performance.

Noise in real-world RGB images is spatially correlated due to the demosaic operation in image signal processing (ISP) pipeline. A feasible solution is to break the noisy correlation with pixel-shuffle downsampling (Zhou et al., 2020), then apply BSN to the downsampled images (Lee et al., 2022; Wang et al., 2023). Apart from that, CVF-SID (Neshatavar et al., 2022) learns a cyclic function to decompose the noisy image into clean and noisy components. (Li et al., 2023) detects flat as well as textured areas, then constructs supervisions for them separately.

## 2.3 Image Restoration Network

Since the pioneering works (Dong et al., 2015; Zhang et al., 2017a), data-driven CNN has been dominantly investigated in image restoration (Lai et al., 2017; Tai et al., 2017; Zhang et al., 2017b; 2018a; Guo et al., 2019; Ren et al., 2019; Abdelhamed et al., 2020; Zhang et al., 2022). Encoder-decoder based U-Net architectures (Nah et al., 2017; Yue et al., 2019; Abuolaim & Brown, 2020; Zhang et al., 2021b; Cho et al., 2021; Zamir et al., 2021) learn hierarchical multi-scale representation while maintaining computational efficiency. Other architecture designs, such as residual block (Kim et al., 2016), dense block (Wang et al., 2018; Zhang et al., 2020), and channel attention (Zhang et al., 2018b; Dai et al., 2019; Anwar & Barnes, 2019) are also proposed to improve the model ability. Recently, transformer (Vaswani et al., 2017) has shown much popularity in computer vision, and has also been introduced for image restoration (Chen et al., 2021; Liang et al., 2021; Li et al., 2021). Incorporating with hierarchical design (Wang et al., 2022b; Zamir et al., 2022), they make better trade-offs between performance and efficiency. In addition, HAT (Chen et al., 2023) shows hybrid design could benefit from the complementary of CNN and transformer. NAFNet (Chen et al., 2022) proposes a simple yet effective network without nonlinear activations.

## 3 Method

Here we first analyze the building concepts of some typical BSNs, *i.e.*, FC-AIDE (Cha & Moon, 2019), Laine19 (Laine et al., 2019), DBSN (Wu et al., 2020) and FBI-Net (Byun et al., 2021). Then we introduce how to adapt advanced image restoration designs into various blind-spot ones.

## 3.1 Building Concepts of Existing Blind-Spot Networks

Blind-spot networks (BSN) aim to exclude the corresponding input pixel from the receptive field of every output position. Existing BSN can mainly be divided into two categories. One category (Cha & Moon, 2019; Laine et al., 2019) implements BSN with multiple network branches that each branch has its receptive field restricted in a specific direction. At the end of the network branches, the features are fused to conduct full receptive field except for blind-spot. The other (Wu et al., 2020; Byun et al., 2021) adopts masked and dilated convolutions. The receptive field of the layers are

Figure 1: Illustration of four typical blind-spot networks. White squares indicate masked pixels and blue ones indicate activated pixels. The upper sub-figure represents the convolution layer settings, and the lower one represents the receptive field of a certain pixel. Please zoom in for details.

particularly designed to maintain the blind-spot mechanism through the whole network. Fig. 1 illustrates network layer settings and receptive fields of the four representative networks. Below we revisit their details.

**FC-AIDE** (Cha & Moon, 2019) applies three network branches with respective field restricted to upper right, upper left and downward, respectively. The restriction of receptive field is implemented by applying mask to the convolution kernels. The output features of three branches are combined with $1 \times 1$ convolutions to create complete receptive field except for blind-spot.

**Laine19** (Laine et al., 2019) also applies multiple network branches with receptive field restricted to different directions. In practice, it inputs noisy images rotated in different directions into one shared branch to improve the parameter efficiency. And U-Net (Ronneberger et al., 2015) architecture is adopted for further computation efficiency.

**DBSN** (Wu et al., 2020) utilizes dilated convolution layers to design blind-spot network. Dilated convolutions have sparse receptive fields whose distance is an integer multiple of the dilation rate. This property maintains when multiple layers with the same dilation rate are stacked. In practice, DBSN adopts $3 \times 3$ center masked convolution at the first layer, and $3 \times 3$ dilated convolutions with dilation rate 2 in the following layers. As a result, in the deeper layers, the receptive field is maintained at positions that are an even number of pixels away from the current pixel. And the center masked convolution at first layer can reverse the receptive field, thus implementing the blind-spot mechanism. Besides, DBSN employs two branches with varying dilation rates to address the issue of an excessive number of missing positions in the receptive field.

**FBINet** (Byun et al., 2021) also applies dilated convolutions to construct BSN. The initial layer adopts handcrafted kernel mask to decrease the redundant blind-spots. In the deep layers, the dilation rate is set to 3 for fast expansion of the receptive field.

## 3.2 ADAPTING IMAGE RESTORATION ARCHITECTURES INTO BLIND-SPOT ONES

In this subsection, we demonstrate how to adapt advanced image restoration architectures (*i.e.*, SwinIR (Liang et al., 2021), Restormer (Zamir et al., 2022), NAFNet (Chen et al., 2022), HAT (Chen et al., 2023)) into various blind-spot ones. Actually, the building concepts of existing BSNs in Sec. 3.1 have provided us some guidelines. On the one hand, for BSN based on multi-branch design (Cha & Moon, 2019; Laine et al., 2019), we first duplicate the image restoration network into multiple branches and restrict the receptive field of each branch to a specific direction. Then we add $1 \times 1$ convolution layers at the end to fuse the features from different branches. On the other hand, for BSN based on dilated convolutions (Wu et al., 2020; Byun et al., 2021), we take masked convolution as the first layer, and turn the deep layers in restoration networks into dilated ones.

However, the novel components (*e.g.*, window attention and channel attention) in image restoration architectures are not considered in existing blind-spot networks. We show how to adapt these components into blind-spot ones in the following.

**Window Attention** enables interactions between contexts within local windows and shows favorable performance against CNN in image restoration (Liang et al., 2021; Chen et al., 2023). Denote the input feature by $\mathbf{X} \in \mathbb{R}^{H \times W \times C}$. It is first partitioned into $\frac{HW}{M^2}$ local windows of size $M \times M$. Within each local window, the feature is projected to *query*, *key* and *value* as $\mathbf{Q}, \mathbf{K}, \mathbf{V} \in \mathbb{R}^{M^2 \times d}$, respectively. Then window-based self-attention can be formulated as,

$$Attention(\mathbf{Q}, \mathbf{K}, \mathbf{V}) = SoftMax(\mathbf{Q}\mathbf{K}^T / \sqrt{d} + \mathbf{B})\mathbf{V}, \tag{1}$$

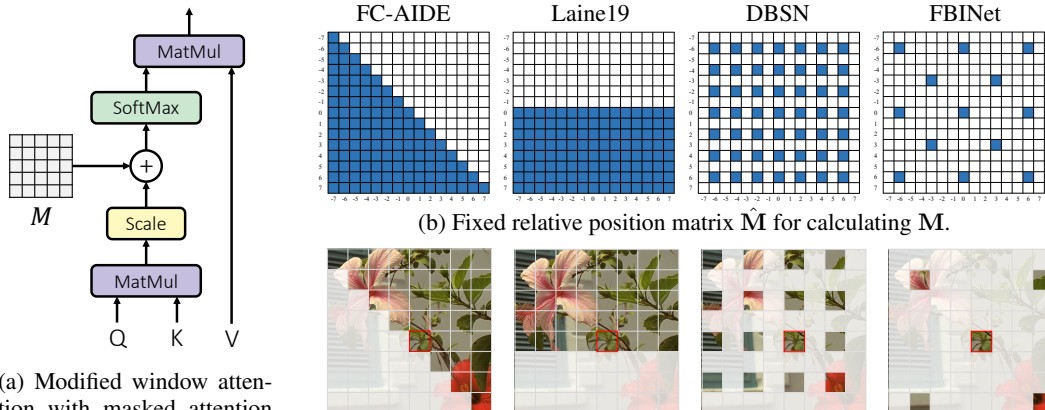

(a) Modified window attention with masked attention weights.

(b) Fixed relative position matrix $\hat{\mathbf{M}}$ for calculating $\mathbf{M}$.

(c) Receptive field of modified window attention.

Figure 2: Adapting window attention into blind-spot ones. (a) shows the window attention with attention mask $\mathbf{M}$, which is to mimic the masked and dilated convolutions. (b) shows the fixed binary relative position matrix $\hat{\mathbf{M}}$ of various blind-spot manners for calculating $\mathbf{M}$. The white squares are set to 0 to mask weights and blue ones are set to 1 to activate weights. (c) shows the receptive field of modified window attention, where white squares represent the blocked areas in the receptive field of the pixel in red box.

where $\mathbf{B} \in \mathbb{R}^{M^2 \times M^2}$ is the learnable relative positional bias, $d$ is the feature dimension. Due to the relative position along each axis lies in the range $[-M+1, M-1]$, $\mathbf{B}$ is parameterized as a smaller bias matrix $\hat{\mathbf{B}} \in \mathbb{R}^{(2M-1) \times (2M-1)}$. And $\mathbf{B}$ is calculated from $\hat{\mathbf{B}}$ based on the relative position, *i.e.*,

$$\mathbf{B}(i, j) = \hat{\mathbf{B}}(x_i - x_j, y_i - y_j), \tag{2}$$

where $i$ and $j$ are two positions within the local window, $(x_i, y_i)$ and $(x_j, y_j)$ are their coordinates along $H$ and $W$ axis, respectively.

In order to implement the same functionality as masked (Cha & Moon, 2019; Laine et al., 2019) or dilated (Wu et al., 2020; Byun et al., 2021) convolution layers, we restrict the receptive field of window attention by masking out certain positions of the attention matrix. Specifically, we introduce a fixed mask $\mathbf{M} \in \mathbb{R}^{M^2 \times M^2}$ to the attention matrix, as shown in Fig. 2 (a). Thus, Eqn. (1) can be modified as,

$$Attention(\mathbf{Q}, \mathbf{K}, \mathbf{V}) = SoftMax(\mathbf{Q}\mathbf{K}^T/\sqrt{d} + \mathbf{B} + \mathbf{M})\mathbf{V}, \tag{3}$$

where $\mathbf{M}$ is a two-value matrix that some locations are set to $-\infty$ to mask attention weights and others are set to 0 to activate weights. Similar to the relative positional bias $\mathbf{B}$, $\mathbf{M}$ is calculated from a fixed binary relative position matrix $\hat{\mathbf{M}} \in \mathbb{R}^{(2W-1) \times (2W-1)}$, *i.e.*,

$$\mathbf{M}(i, j) = \begin{cases} 0, & \text{if } \hat{\mathbf{M}}(x_i - x_j, y_i - y_j) = 1 \\ -\infty, & \text{if } \hat{\mathbf{M}}(x_i - x_j, y_i - y_j) = 0 \end{cases}. \tag{4}$$

The relative position matrix $\hat{\mathbf{M}}$ for adapting various BSN is shown in Fig. 2 (b). From Fig. 2 (c), the receptive field of our modified window attention is identity with dilated or masked convolutions.

In addition, we also noticed that SwinIA (Papkov & Chizhov, 2023) implements BSN with window attention as well. It initializes query features with positional encoding, and takes shallow features of the noisy input as key and value features. Associated with an attention mask, the query features do not interact with the key and value features at the same location, thereby achieving the blind-spot mechanism. However, this implementation shows inferior performance to the convolutional counterparts such as (Laine et al., 2019). Instead, we mimic the layers in convolution-based BSNs with masked window attention to overcome its limitation.

**Channel Attention** enables global interaction and has been widely adopted in state-of-the-art image restoration architectures (Chen et al., 2022; 2023). Given an input feature $\mathbf{X} \in \mathbb{R}^{H \times W \times C}$, channel attention $CA$ can be formalized as,

$$CA(\mathbf{X}) = \mathbf{X} * \phi(\mathbf{X}), \tag{5}$$

Table 1: Ablation study on channel attention.

| Laine19 | Baseline | +Restormer | +NAFNet |
|---------|----------|------------|---------|
| w/ CA | - | 30.12 / 0.788 | 22.60 / 0.595 |
| w/o CA | 32.26 / 0.881 | 32.52 / 0.884 | 32.68 / 0.888 |

where $*$ is a channel-wise product operation. The function $\phi$ should aggregate spatial information of each channel and output a vector with size $1 \times 1 \times C$. For example, NAFNet (Chen et al., 2022) achieves $\phi$ by global average pooling and a linear layer, and Restormer (Zamir et al., 2022) utilizes self-attention operations in the channel dimension to extract global information.

Note that BSN needs to meet the requirement that excepts the corresponding input pixel from the receptive field of an output pixel. However, $\phi(\mathbf{X})$ aggregates the information of all spatial locations and has the global receptive field, which may violate the blind-spot requirement. Actually, in experiments, we found that whether it is harmful to SSID depends on the spatial size and channel width. In single-scale architectures (*e.g.*, HAT (Chen et al., 2023)) the spatial size is maintained constant through the network layers and it is much larger than the channel number. Thus, the spatial information is largely compressed by $\phi(\mathbf{X})$, making it safe to adopt CA. However, when it turns to hierarchical multi-level architectures (Zamir et al., 2022; Chen et al., 2022), the features are downsampled multiple times at the deep levels. Thus, the spatial information may be distributed in channel dimensions, which may cause information leaks of the blind-shot values. In order to avoid this negative effect, we remove the channel attention in (Zamir et al., 2022; Chen et al., 2022) when implementing BSN with hierarchical architecture (Laine et al., 2019).

**Downsampling and Upsampling** are commonly associated with hierarchical architectures in image restoration (Zamir et al., 2022; Chen et al., 2022). They process in multi-scale resolutions and are computationally efficient. However, several existing BSNs are single-scale architectures where the feature resolution is maintained the same as that of input images through the network. When adapting to these types of BSNs, downsampling and upsampling operations may conflict with the blind-spot mechanism. For simplicity, we remove the downsampling and upsampling operations in image restoration architectures when implementing BSNs with straight forward architectures (Cha & Moon, 2019; Wu et al., 2020; Byun et al., 2021).

## 4 EXPERIMENTS

### 4.1 IMPLEMENTATION DETAILS

**Datasets for Synthetic Denoising**. Following Laine19 (Laine et al., 2019), our training data contains 44328 images of size between $256 \times 256$ and $512 \times 512$ from ImageNet (Deng et al., 2009) validation dataset. Our test datasets are commonly used Kodak (24 images), BSD300 validation set (100 images), and Set14 (14 images). We consider four synthetic noise types: Gaussian noise of $\sigma = 25$, Gaussian noise of $\sigma \in [5, 50]$, Poisson noise of $\lambda = 30$, Poisson noise of $\lambda \in [5, 50]$.

**Datasets for Real-World Denoising**. Smartphone image denoising dataset (SIDD) (Abdelhamed et al., 2018) collects noisy-clean pairs from five smartphone cameras. Each noisy image is captured multiple times and the average image is served as ground truth. It provides 320 training pairs (SIDD-Medium) and 40 testing images, while 1280 validation patches and 1280 benchmark patches are cropped from the testing images. We train our network on SIDD-Medium dataset and test on the validation and benchmark patches. Darmstadt noise dataset (DND) (Plotz & Roth, 2017) is a test set captured from DSLR camera. The noisy image is captured with a short exposure time while the corresponding clean image is captured with a long exposure time. It contains 50 noisy images for test only. We train and test our method on the test images in a fully self-supervised manner.

**Training Details**. Some training settings are the same for synthetic and real-world noise. We crop the training images into patches of size $128 \times 128$ to facilitate the network training. We use Adam (Kingma & Ba, 2014) optimizer. The batch size is set to 8 and the initial learning rate is set to $3 \times 10^{-4}$. For synthetic noise, we train the network with negative log-likelihood loss and test with posterior inference (Laine et al., 2019). The learning rate is decreased to zero with cosine annealing scheduler (Loshchilov & Hutter, 2016), and the network is trained total 500k iterations. For real-world noise, we apply asymmetric pixel shuffle down-sampling rate for training and inference, and improve the results with random replacement refinement (R3) (Lee et al., 2022). We use L1 loss

Table 2: Ablation study on synthetic Gaussian denoising of $\sigma = 25$. PSNR (dB) and SSIM are measured on Kodak24 dataset. The best results are shown in **bold**.

|  | Noise2Void | NBR2NBR | FC-AIDE | Laine19 | DBSN | FBINet |
|---|---|---|---|---|---|---|
| Baseline | 30.07 / 0.818 | 32.08 / 0.879 | 32.21 / 0.878 | 32.26 / 0.881 | 32.18 / 0.876 | 32.33 / 0.882 |
| +SwinIR | 30.96 / 0.843 | 32.25 / 0.880 | 32.36 / 0.880 | 32.58 / 0.886 | 32.34 / 0.879 | 32.34 / 0.882 |
| +Restormer | **30.99 / 0.846** | 32.29 / 0.881 | 32.38 / 0.881 | 32.52 / 0.884 | 32.46 / 0.881 | 32.41 / 0.883 |
| +HAT | **30.99** / 0.844 | **32.41 / 0.883** | **32.43 / 0.882** | 32.66 / 0.887 | **32.47** / 0.881 | 32.35 / 0.882 |
| +NAFNet | 30.96 / **0.846** | 32.30 / 0.882 | 32.39 / 0.881 | **32.68 / 0.888** | **32.47 / 0.882** | **32.56 / 0.886** |

Table 3: Ablation study on real-world denoising. PSNR (dB) is measured on SIDD validation dataset.

|  | FC-AIDE | Laine19 | DBSN | FBINet |
|---|---|---|---|---|
| Baseline | 36.89 | 36.79 | 37.02 | 36.98 |
| +SwinIR | 36.87 | 36.97 | 37.12 | **37.14** |
| +Restormer | **37.10** | 37.16 | 37.09 | 37.10 |
| +HAT | 36.90 | 37.01 | 37.01 | 37.06 |
| +NAFNet | 36.95 | **37.20** | **37.28** | 37.07 |

Table 4: Ablation study on NAFNet on SIDD validation dataset.

| DConv | LN | SCA | Activation | PSNR |
|---|---|---|---|---|
| ✓ | ✓ | ✓ | Simple Gate | 37.28 |
| ✓ | ✓ | ✓ | ReLU | 37.31 |
| ✓ | ✓ | ✗ | ReLU | 37.30 |
| ✓ | ✗ | ✗ | ReLU | **37.42** |
| ✗ | ✗ | ✗ | ReLU | 37.32 |

to train the network, and the learning rate is decreased by 10 every 40k iterations with total 100k iterations. The experiments are conducted on Nvidia Tesla V100 GPUs.

The BSNs and image restoration architectures are varied in network size and computation cost. For a fair comparison, we adjust the channel numbers and network depth to control the computation cost of all the methods to be the same (*i.e.*, 100G FLOPs @ $256 \times 256$) in our ablation study. We scale up to larger models (*i.e.*, $\tilde{4}$00G FLOPs @ $256 \times 256$) when comparing to state-of-the-art methods, denoted as Ours-L.

## 4.2 ABLATION STUDY

**Ablation study on channel attention**. In Sec. 3.2, we demonstrate that channel attention may leak the information when adapting image restoration networks (*i.e.*, Restormer (Zamir et al., 2022) and NAFNet (Chen et al., 2022)) into hierarchical BSN (Laine et al., 2019). As shown in Tab. 1, in such situations, channel attention leads to an obvious performance drop, which seems to be overfitted to the noisy images. Instead, large improvements are achieved by removing the channel attention.

**Ablation study on synthetic noise**. We adapt four representative image restoration architectures into blind-spot ones according to Sec. 3.2 to assess their effectiveness in BSN-based SSID methods. In addition, for SSID methods (Krull et al., 2019; Laine et al., 2019) adopting normal networks that implement blind-spot mechanism with masked inputs, we simply replace their networks with image restoration architectures. From Tab. 2, all the restoration networks bring noticeable improvements over the baselines, which encourages to adoption of advanced architecture designs in SSID methods. Among the image restoration architectures, NAFNet (Chen et al., 2022) achieves the largest average improvement, which is consistent with the conclusion in image restoration. The BSN-based methods show better overall performance than mask-based ones (Krull et al., 2019; Huang et al., 2021), while Laine19 (Laine et al., 2019) perform best on SSID. This may be because the work (Laine et al., 2019) has a complete receptive field (except for blind-spot) and hierarchical architecture. In conclusion, adapting NAFNet into Laine19 (Laine et al., 2019) shows the best results on synthetic noise. We adopt this implementation for synthetic denoising.

**Ablation study on real-world noise**. As shown in Tab. 3, implementing BSNs with advanced image restoration architectures also shows large improvements on real-world denoising. Different from synthetic noise, DBSN shows the best performance among BSNs, which is consistent with the conclusion of AP-BPN (Lee et al., 2022). It may be because the dilated convolutions in DBSN have sparse receptive fields similar to pixel-shuffle downsampling (PD). As AP-BPN (Lee et al., 2022) adopts different PD factors between training and inference, DBSN may have better generalization ability from training PD factor to inference. In addition, although NAFNet has shown good performance, we empirically found that novel components in NAFNet may cause unstable training in SSID, leading to limited performance. Tab. 4 analysis the basic components, *e.g.*, Depth-wise Convolution (DConv), Layer Normalization (LN), simplified channel attention (SCA) and activa-

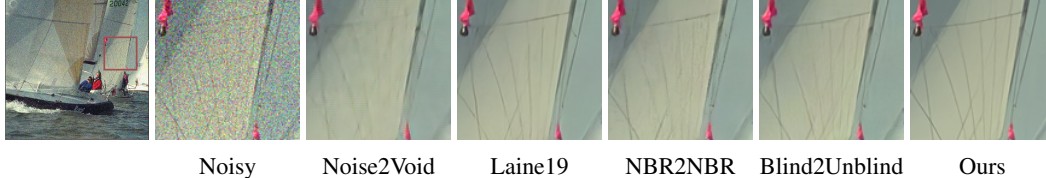

| Noisy | Noise2Void | Laine19 | NBR2NBR | Blind2Unblind | Ours |

Figure 3: Qualitative comparison on Gaussian denoising of $\sigma = 25$ on Kodak dataset.

Table 5: Quantitative comparison on synthetic denoising. PSNR (dB) and SSIM are measured on Kodak dataset. Results on BSD300 and Set14 datasets are provided in the appendix.

| Method | Gaussian $\sigma = 25$ | Gaussian $\sigma \in [5, 50]$ | Poisson $\lambda = 30$ | Poisson $\lambda \in [5, 50]$ |
|---|---|---|---|---|
| Noise2Void (2019) | 30.32 / 0.821 | 30.32 / 0.821 | 28.90 / 0.788 | 28.78 / 0.758 |
| Laine19 (2019) | 32.40 / 0.883 | 32.40 / 0.870 | 31.67 / 0.874 | 30.88 / 0.850 |
| Self2Self (2020) | 31.28 / 0.864 | 31.37 / 0.860 | 30.31 / 0.857 | 29.06 / 0.834 |
| DBSN (2020) | 31.64 / 0.856 | 30.38 / 0.826 | 30.38 / 0.826 | 29.60 / 0.811 |
| R2R (2021) | 32.25 / 0.880 | 31.50 / 0.850 | 30.50 / 0.801 | 29.14 / 0.732 |
| NBR2NBR (2021) | 32.08 / 0.879 | 32.10 / 0.870 | 31.44 / 0.870 | 30.86 / 0.855 |
| Blind2Unblind (2022) | 32.27 / 0.880 | 32.34 / 0.872 | 31.64 / 0.871 | 31.07 / 0.857 |
| Ours | **32.67 / 0.888** | **32.54 / 0.886** | **31.86 / 0.877** | **31.40 / 0.862** |
| Ours-L | **32.81 / 0.890** | **32.68 / 0.888** | **31.98 / 0.879** | **31.61 / 0.870** |

tion function in the building block of NAFNet. Removing some of the components (*i.e.*, LN and SCA) and replacing the simple gate activation function with ReLU provide a more stable training process and slightly better performance. We take this implementation for real-world denoising.

### 4.3 RESULTS FOR SYNTHETIC DENOISING

Benefiting from advanced architectures, our base model with 100G FLOPs computation cost achieves favorable performance against state-of-the-art SSID methods, which could be further improved with larger model size. The quantitative results of synthetic denoising on Kodak dataset are shown in Tab. 5, results on BSD300 and Set14 datasets are provided in the appendix. Noise2Void (Krull et al., 2019) can not sufficiently restore the clean signal due to the information loss at blind-spot. Laine19 (Laine et al., 2019) mitigates this problem with Bayesian inference, which has the potential to fully recover the clean information. In addition, R2R (Quan et al., 2020) synthesis training pairs from single noisy images for network training. Neighbor2Neighbor (Huang et al., 2021) and Blind2Unblind (Wang et al., 2022a) apply regular loss functions to add back the information in the blind-spot. These methods exhibit performance close to their supervised counterparts. Nevertheless, all the above methods are based on U-Net or plain convolutional architecture, which largely limits their modeling ability. Equipped with advanced neural architecture in image restoration, we achieved up to 0.54dB improvement in terms of PSNR. This demonstrates the effectiveness of keeping up with advanced networks in image restoration. Fig. 3 shows visual comparisons of synthetic denoising, where our method sufficiently removes the noise and preserves the fine-grained details on the sailboat.

### 4.4 RESULTS FOR REAL-WORLD DENOISING

Quantitative results for real-world denoising are shown in Tab. 6. Blind-spot techniques designed for synthetic noise (Krull et al., 2019; Batson & Royer, 2019) assume the noise is spatial independent, which shows little denoising effect on real-world noisy images. DBSN (Wu et al., 2020) firstly applies pixel-shuffle downsampling to break the noise correlation, then denoises the noisy image with blind-spot network. AP-BSN (Lee et al., 2022) introduces asymmetric PD factors to trade-off between texture details and noise removal. However, their performance is largely limited by the plain convolutional BSN architecture. Recently, LG-BPN (Wang et al., 2023) overcomes this limitation by incorporating with transformer block (Zamir et al., 2022) for global information. Instead, our BSN adapted from advanced image restoration architecture shows 1.06dB improvement against AP-BSN on SIDD validation dataset, while also surpassing the state-of-the-art methods by a large margin.

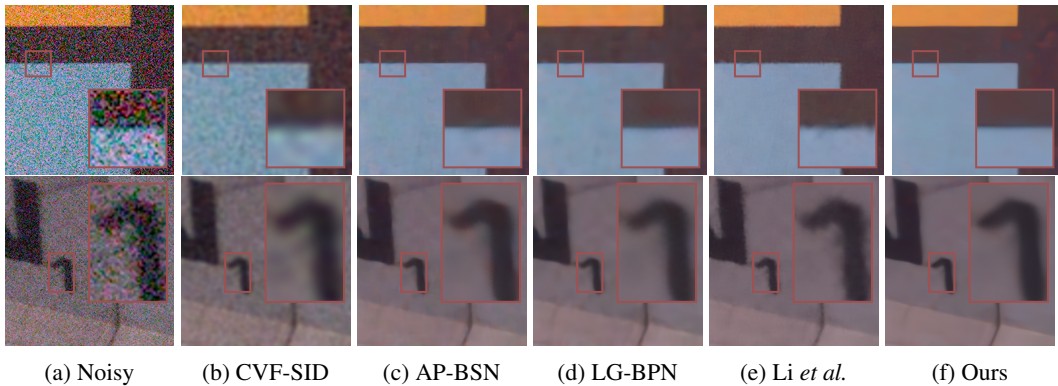

(a) Noisy      (b) CVF-SID      (c) AP-BSN      (d) LG-BPN      (e) Li *et al.*      (f) Ours

Figure 4: Qualitative comparison on SIDD dataset (Abdelhamed et al., 2018).

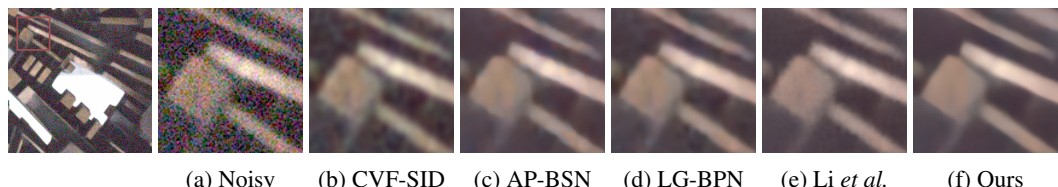

(a) Noisy      (b) CVF-SID      (c) AP-BSN      (d) LG-BPN      (e) Li *et al.*      (f) Ours

Figure 5: Qualitative comparison on DND dataset (Plotz & Roth, 2017).

Table 6: Quantitative comparison on real-world denoising. PSNR (dB) and SSIM are measured on SIDD and DND datasets. Due to DND benchmark website is down before paper submission, we will update the results as soon as the website is recovered.

|  | Method | SIDD Validation | SIDD Benchmark | DND Benchmark |
|---|---|---|---|---|
| Unpaired | GCBD (2018) | - | - | 35.58 / 0.922 |
|  | UIDNet (2020) | - | 32.48 / 0.897 | - |
|  | Wu *et al.* (2020) | - | - | 37.93 / 0.937 |
|  | C2N (2021) | 35.36 / 0.932 | 35.35 / 0.937 | 37.28 / 0.924 |
| Self-Supervised | Noise2Void (2019) | 27.48 / 0.664 | 27.68 / 0.668 | - |
|  | Noise2Self (2019) | 29.94 / 0.782 | 29.56 / 0.808 | - |
|  | NAC (2020) | - | - | 36.20 / 0.925 |
|  | R2R (2021) | - | 34.78 / 0.898 | - |
|  | CVF-SID (2022) | 34.15 / 0.911 | 34.71 / 0.917 | 36.50 / 0.924 |
|  | AP-BSN (2022) | 36.74 / 0.934 | 36.91 / 0.931 | 38.09 / 0.937 |
|  | Li *et al.* (2023) | 37.39 / 0.934 | 37.41 / 0.934 | 38.18 / 0.938 |
|  | LG-BPN (2023) | - | 37.28 / 0.936 | 38.43 / 0.942 |
|  | Ours | **37.42 / 0.935** | **37.36 / 0.934** | - |
|  | Ours-L | **37.80 / 0.940** | **37.74 / 0.939** | - |

Fig. 4 and Fig. 5 show visual comparisons in real-world scenarios. Our model can remove the spatial correlated noise smoothly.

## 5   CONCLUSION

In this paper, we notice the lack of research on neural architectures in self-supervised image denoising (SSID). We suggest adopting the advanced designs in image restoration and introduce a series of approaches to adapt them into SSID. Specifically, we propose a fixed attention mask based on the relative position for window attention to mimic the convolutional counterparts. And we discourage the use of channel attention in hierarchical architectures. Experiments on representative restoration networks migrating various blind-spot networks show consistent improvement against the convolutional baselines. We hope our study could bring attention to the development of more advanced BSN architectures, and serve as a foundational reference for future works.

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

APPENDIX

Here we provide the quantitative results of synthetic denoising on BSD300 and Set14 datasets in Tab. A and Tab. B, respectively.

Table A: Quantitative comparison of synthetic image denoising on BSD300 dataset.

| Method | Gaussian $\sigma = 25$ | Gaussian $\sigma \in [5, 50]$ | Poisson $\lambda = 30$ | Poisson $\lambda \in [5, 50]$ |
|---|---|---|---|---|
| Noise2Void (2019) | 29.34 / 0.824 | 29.31 / 0.801 | 28.46 / 0.798 | 27.92 / 0.766 |
| Laine19 (2019) | 30.99 / 0.877 | 30.95 / 0.861 | 30.25 / 0.866 | 29.57 / 0.841 |
| Self2Self (2020) | 29.86 / 0.849 | 29.87 / 0.841 | 28.93 / 0.840 | 28.15 / 0.817 |
| DBSN (2020) | 29.80 / 0.839 | 28.34 / 0.788 | 28.19 / 0.790 | 27.81 / 0.771 |
| R2R (2021) | 30.91 / 0.872 | 30.56 / 0.855 | 29.47 / 0.811 | 28.68 / 0.771 |
| NBR2NBR (2021) | 30.79 / 0.873 | 30.73 / 0.861 | 30.10 / 0.863 | 29.54 / 0.843 |
| Blind2Unblind (2022) | 30.87 / 0.872 | 30.86 / 0.861 | 30.25 / 0.862 | 29.92 / 0.852 |
| Ours | **31.29 / 0.884** | **31.20 / 0.880** | **30.42 / 0.871** | **30.18 / 0.861** |
| Ours-L | **31.39 / 0.886** | **31.32 / 0.884** | **30.60 / 0.874** | **30.34 / 0.867** |

Table B: Quantitative comparison of synthetic image denoising on Set14 dataset.

| Method | Gaussian $\sigma = 25$ | Gaussian $\sigma \in [5, 50]$ | Poisson $\lambda = 30$ | Poisson $\lambda \in [5, 50]$ |
|---|---|---|---|---|
| Noise2Void (2019) | 28.84 / 0.802 | 29.01 / 0.792 | 27.73 / 0.774 | 27.43 / 0.745 |
| Laine19 (2019) | 31.36 / 0.866 | 31.21 / 0.855 | 30.47 / 0.855 | 28.65 / 0.785 |
| Self2Self (2020) | 30.08 / 0.839 | 29.97 / 0.849 | 28.84 / 0.839 | 28.83 / 0.841 |
| DBSN (2020) | 30.63 / 0.846 | 29.49 / 0.814 | 29.16 / 0.814 | 28.72 / 0.800 |
| R2R (2021) | 31.32 / 0.865 | 30.84 / 0.850 | 29.53 / 0.801 | 28.77 / 0.765 |
| NBR2NBR (2021) | 31.09 / 0.864 | 31.05 / 0.858 | 30.29 / 0.853 | 29.79 / 0.838 |
| Blind2Unblind (2022) | 31.27 / 0.864 | 31.14 / 0.857 | 30.46 / 0.852 | 30.10 / 0.844 |
| Ours | **31.47 / 0.872** | **31.42 / 0.869** | **30.70 / 0.858** | **30.31 / 0.856** |
| Ours-L | **31.58 / 0.874** | **31.51 / 0.872** | **30.88 / 0.860** | **30.44 / 0.858** |

