# OpenReview forum: "Towards Bringing Advanced Restoration Networks into Self-Supervised Image Denoising"
_ICLR.cc/2024/Conference — ICLR 2024 Conference Withdrawn Submission_

### Official Review · Reviewer_jHRD · 2023-10-29

**Soundness:** 2 fair
**Presentation:** 2 fair
**Contribution:** 2 fair
**Rating:** 5
**Confidence:** 3

**Summary:**

The main idea of this article is to enhance the performance of self-supervised image denoising (SSID) by integrating advanced image restoration network designs into SSID methods. Traditional SSID approaches often rely on simple neural network architectures, which have become outdated in light of recent developments in image restoration networks. The authors aim to bridge this gap by adapting advanced network architectures and attention mechanisms from image restoration to SSID. Experimental results demonstrate performance improvements on synthetic and real-world noisy images.

**Strengths:**

1. The paper delves into the relationship between network architectures and SSID methods.
2. The paper proposes several techniques to improve the network architecture of SSID.
2. The paper demonstrates the achievement of more advanced and improved results in the field of SSID.

**Weaknesses:**

1. The paper appears to lack significant innovation as the consensus in the field is that using more advanced and complex network architectures to enhance performance is well-established.
2. While the paper introduces some modifications to network structures to make them suitable for SSID, these changes are relatively minor, such as the introduction of channel attention and other attention mechanisms.
3. It appears that the network modifications may not yield as significant improvements as directly increasing the network parameters would. It would be informative to investigate the results of directly increasing the parameters of the original U-Net network. Ultimately, as previously mentioned, whether through improving the network or increasing parameters, the performance improvements seem somewhat expected in the context of prior neural network-based work.

**Questions:**

See weakness

---

### Official Review · Reviewer_iVws · 2023-10-31

**Soundness:** 3 good
**Presentation:** 3 good
**Contribution:** 1 poor
**Rating:** 3
**Confidence:** 4

**Summary:**

This paper aims to adapt transformer based models for self-supervised image denoising. To achieve the blind spot effect, the authors start by analyzing the previous blind spot mechanism in detail. Inspired by the mechanism in previous works, the authors proposed the blind spot window attention by adjusting the masks. The channel attention, downsampling, and upsampling blocks are difficult to deal with. And it is quite likely that information leaks will happen in the network. To avoid that, the authors simply remove the channel attention, downsampling, and upsampling operations in the network. Experiments are conducted on various datasets.

**Strengths:**

1. The paper is well-written and easy to understand.

2. At the beginning of the paper, the paper talks about how the blind-spot mechanism could be achieved by analyzing the four blind-spot networks. And this analysis leads naturally to the extension of window self-attention.

3. The authors conducted various experiments and provided sufficient experimental results to support the effectiveness of the proposed mechanism.

**Weaknesses:**

1. The main concern is the novelty of this paper. There are mainly three parts in a transformer network that might influence the blind-spot mechanism. The first one is window self-attention and it is adapted to  blind-spot networks naturally. The other two operations includes channel attention, downsampling and upsampling. Yet, the possible information leak is only solved by discarding those components. This is quite a brute-force approach.

2. The biggest of removing the downsampling and upsampling operations enforce that all operation is done on the same resolution as the input image. Yet, this corresponds to a significant increase of the computation. Thus, it becomes questionable whether it is good to have a lightly improved performance by introducing too much computation.

**Questions:**

1. In network such as HAT, and swinir. there are 3x3 convolutions in the network. How the blind-spot mechanism of those 3x3 convolution kernels are guaranteed.

2. Are there experiments  on synthetic dataset?

---

### Official Review · Reviewer_uBSH · 2023-11-01

**Soundness:** 2 fair
**Presentation:** 3 good
**Contribution:** 1 poor
**Rating:** 3
**Confidence:** 5

**Summary:**

This work investigated how to incorporate transformers into blind-spot based self-supervised image denoising (SSID) by adjusting window attention and not using channel attention. This work investigated various SSID methods with different blind-spot strategies and demonstrated its great denoising performance in both synthetic and real denoising tasks.

**Strengths:**

Reporting state-of-the-art performance in popular synthetic and real benchmarks is promising.
Considering different blind-spot strategies in self-supervised denoising is neat.

**Weaknesses:**

Unlike the claim in this manuscript, self-supervised image denoising with transformers has been investigated in multiple works. It is unclear if the proposed method is novel over these prior works.
The motivation of this work ("We notice the lack of research on backbone architectures in SSID and suggest adapting the advanced designs in restoration networks into SSID") seems unclear - why transformers should work better than CNNs in denoising?

**Questions:**

Q1. here are prior works on self-supervised denoising with transformers. Please clarify the novelty of the proposed method over them as well as demonstrate its superiority to them.
- Young-Joo Han, Ha-Jin Yu, SS-BSN: Attentive Blind-Spot Network for Self-Supervised Denoising with Nonlocal Self-Similarity, IJCAI 2023
- X Liu et al., DnT: Learning Unsupervised Denoising Transformer from Single Noisy Image, IPMV 2022 (doi: 10.1145/3529446.3529455)
- LG-BPN (Wang et al., 2023), which was cited in the manuscript, but need more explanation.

Q2. the motivation of this work is simply mentioning "we notice the lack of research on backbone architectures in SSID and suggest adapting the advanced designs in restoration networks into SSID," but it is unclear why transformers should work better than CNNs in denoising. CNN still works well in very recent works like the following works, so strong justification of using transformers in denoising may be needed.
- J Li et al., Spatially Adaptive Self-Supervised Learning for Real-World Image Denoising, CVPR 2023
- D Zhang et al., MM-BSN: Self-Supervised Image Denoising for Real-World with Multi-Mask based on Blind-Spot Network, CVPRW 2023
- Y Zou et al., Iterative Denoiser and Noise Estimator for Self-Supervised Image Denoising, ICCV 2023
- J Wang et al., Noise2Info: Noisy Image to Information of Noise for Self-Supervised Image Denoising, ICCV 2023

Q3. The proposed method may have limited extension capability. Recent works on self-supervised single image denoising now uses very simple and lightweight CNN networks (not even U-net) to achieve remarkable results (see the below works) while it is unclear transformer-based networks can achieve similar performance in denoising considering that transformers usually require larger dataset than CNNs. I am afraid that this work goes into the opposite direction of using heavier network for denoising. Please discuss.
- Y Mansour and R Heckel, Zero-Shot Noise2Noise: Efficient Image Denoising without any Data, CVPR 2023
- J Lequyer et al., A fast blind zero-shot denoiser, Nature Machine Intelligence, 2022.

Q4. Tables 5 and 6 look promising, but there are a number of other information needed such as training time and network size for fair comparisons. Please report.

---

### Official Review · Reviewer_xn1Q · 2023-11-04

**Soundness:** 3 good
**Presentation:** 3 good
**Contribution:** 3 good
**Rating:** 5
**Confidence:** 5

**Summary:**

The paper address self-supervised denoising task, where advanced restoration network designs ( SwinIR, Restormer, NAFNet, and HAT)  are incorporated into blind-spot self-supervised image denoising (SSID) networks. This paper introduce a series of approaches to adapt restoration networks into various blind-spot ones, where they suggest effective adjustment for window attention to mimic the convolution layers in BSN.

**Strengths:**

- The paper conducts the study where networks like SwinIR, Restormer, NAFNet, and HAT can be adapted to BSN networks to perform SSID task
- The paper discusses why channel wise is attention is not beneficial and how it can leaks blind-spot information
- The paper introduced efficient masking strategy to adapt the restoration networks to blind-spot ones
- THe paper performs extensive experiments to on both synthetic and real-world RGB noisy images demonstrate the proposed
methods substantially enhance SSID performance

**Weaknesses:**

- Can authors explain if the proposed can be applicable for both signal depdent and signal independent noises
- Can authors explain if the proposed method can handle other types noises like Poisson, and other types of degradations like chromatic aberration, and jpeg compression etc.
- Can authors explain if the proposed method of experiments can be extended other image restoration tasks like deblurring, adverse weather removal, inpainting
- Can authors show some visualizations of attentions and masks for SSID task and supervised image denoising tasks for atleast one method like SwinIR or Restormer. It would be really helpful for reader to understand how the attentions are being adapted for blind spot ones.
- Can the proposed method experiments be extended to dynamic windowed attentions, or dynamic deformable kernel attentions

**Questions:**

Please refer weaknesses